# *Coniochaeta massiliensis* sp. nov. Isolated from a Clinical Sampl28

**DOI:** 10.3390/jof8100999

**Published:** 2022-09-23

**Authors:** Jihane Kabtani, Muriel Militello, Stéphane Ranque

**Affiliations:** 1Institut Hospitalo-Universitaire Méditerranée Infection, 19-21 Boulevard Jean Moulin, 13005 Marseille, France; 2Faculty of Medical and Paramedical Sciences, Aix-Marseille Université, AP-HM, IRD, SSA, MEPHI, 13005 Marseille, France; 3Department of Mycology and Parasitology, Aix-Marseille Université, AP-HM, IRD, SSA, VITROME, 13005 Marseille, France

**Keywords:** biolog phenotypic technology, *Coniochaeta*, energy-dispersive X-ray spectroscopy, genotype, multilocus DNA sequencing, one new taxon, yeast

## Abstract

The genus *Coniochaeta* belongs to the class Ascomycota and the family Coniochaetaceae. Some of the *Coniochaeta* species are plant and animal pathogens, while others are known to be primarily involved in human diseases. In the last few decades, case reports of human infections with *Coniochaeta* have increased, mainly in immunocompromised hosts. We have described and characterised a new species in the genus *Coniochaeta*, here named *Coniochaeta massiliensis* (PMML0158), which was isolated from a clinical sample. Species identification and thorough description were based on apposite and reliable phylogenetic and phenotypic approaches. The phylogenetic methods included multilocus phylogenetic analyses of four genomic regions: ITS (rRNA Internal Transcribed Spacers 1 and 2), TEF-1α (Translation Elongation Factor-1alpha), B-tub2 (β-tubulin2), and D1/D2 domains (28S large subunit rRNA). The phenotypic characterisation consisted, first, of a physiological analysis using both EDX (energy-dispersive X-ray spectroscopy) and Biolog^TM^ advanced phenotypic technology for fixing the chemical mapping and carbon-source oxidation/assimilation profiles. Afterwards, morphological characteristics were highlighted by optical microscopy and scanning electron microscopy. The in vitro antifungal susceptibility profile was characterised using the E-test^TM^ exponential gradient method. The molecular analysis revealed the genetic distance between the novel species *Coniochaeta massiliensis* (PMML0158) and other known taxa, and the phenotypic analysis confirmed its unique chemical and physiological profile when compared with all other species of this genus.

## 1. Introduction

*Coniochaeta* spp. are pleomorphic ascomycetous fungi belonging to the family *Coniochaetaceae* [1]. Some of these yeasts are also classified within dematiaceous fungi due to the presence of melanin in cell walls, known for emitting dark pigments in culture, which is perceived as a virulence factor [2]. *Coniochaeta* species are ubiquitously distributed in the environment. They have been isolated from several natural substrates, including soil [3,4], wood [5], plants [6], water [7], and food [8]. The previously given name *Coniochaeta* has been retained for the *Lecythophora* genus, following the “one fungus, one name” nomenclature change in January 2013. Therefore, all six species of the *Lecythophora* genus (*L. decumbens*, *L*. *fasciculata*, *L*. *hoffmannii, L*. *lignicola*, *L*. *luteoviridis*, and *L*. *mutabilis*) were transferred to the *Coniochaeta* genus [9,10]. *C. hoffmannii* and *C. mutabilis* are considered the most frequent species of the *Coniochaeta* genus found in clinical samples. These yeasts are known to be human pathogens, causing invasive fungal infections, occasionally with fatal outcomes, particularly in immunocompromised patients [11]. *C. hoffmannii* has been described as a plant pathogen [12]. It has also been implicated in emerging human fungal infections, including subcutaneous abscess [12], keratitis [13], and sinusitis [14], and in canine osteomyelitis in a dog [15]. However, *Coniochaeta mutabilis* [12] is known to be more frequently involved in severe infections. It has been involved in human peritonitis [16], endocarditis [17,18], septic shock [19], endophthalmitis [20,21], and keratomycosis [22]. The present study aimed to properly describe and characterise, genetically and phenotypically, a new species of *Coniochaeta* isolated from a human abscess.

## 2. Materials and Methods

### 2.1. Fungal Strains

The phenotypic features of *Coniochaeta massiliensis* PMML0158 were compared with those of two reference strains, *C. mutabilis* DSM 10716 and *C. hoffmannii* DSM 2693. The genotypic characteristics of these three strains were also compared with those of seven other reference strains in the genus *Coniochaeta*, including *C. fasciculate* CBS 205.38, *C. lignicola* CBS 267.33, *C. luteoviridis* CBS 206.38, *C. hoffmannii* CBS 245.38, *C. mutabilis* CBS 157.44, *C. lignaria* DWS9m2/SMH2569/95.605, and *C. cateniformis* UTHSC 01-1644. *Phialemonium obovatum* CBS 279.76 was used as an outgroup in the phylogenetic analyses.

### 2.2. Macroscopic Characterisation

The temperature growth profiles and macroscopic features, including colony time of growth, aspect, and surface/reverse colour, for all strains were determined with five-day-old colonies cultivated on Sabouraud Dextrose Agar (SDA) plates supplemented with Gentamycin and Chloramphenicol (GC). Colonies were inoculated on other, new plates of SDA GC at 4, 25, 30, 33, 37, 40, and 45 °C.

### 2.3. Microscopic Characterisation

Microscopic features were first analysed with optical microscopy. Microscopic slides were prepared using the cellophane adhesive-tape method with lactophenol cotton blue (LCB). Photographs were taken with the DM 2500 (LEICA CAMERA SARL, Paris, France) camera. Further observation was performed using scanning electron microscopy. A fragment of fungal colonies was fixed on a microscope slide with 400 μL of glutaraldehyde 2.5% in 0.1 M sodium cacodylate buffer and stored at 30 °C for at least 30 min. Photographs were taken with the TM4000 Plus microscope (Hitachi High-Technologies, Tokyo, Japan) adjusted to 15 KeV lens mode 4 with a back-scattered electron detector (BSE).

### 2.4. MALDI-TOF MS Identification

The fungi were incubated on SDA GC at 30 °C for five to eight days. Once grown, colonies were identified with matrix-assisted laser desorption/ionization–time of flight mass spectrometry (MALDI-TOF MS), using the procedure described in Cassagne et al. (2016) [23]. The Microflex LT^TM^ instrument and the MALDI Biotyper^TM^ system (Bruker Daltonics GmbH, Bremen, Germany) were used, along with the manufacturer’s and in-house reference spectra databases, as described in Normand et al. (2017) [24]. 

### 2.5. Antifungal Susceptibility Testing (AFST)

The in vitro sensitivity of nine antifungal drugs was tested against the three *Coniochaeta* strains using the Sensititre YeastOne^TM^ (Thermo Fisher Scientific, Dardilly, France) microdilution system, following the supplier’s recommendations. Briefly, isolates were grown on SDA GC until maturity, the inoculum was suspended in 2 mL of saline, and turbidity was adjusted to 0.5 McFarland, to obtain an inoculum of ~1.5 × 10^8^ CFU/mL. Next, 20 µL of this solution was added to 10 mL of YeastBroth^TM^ (Thermo Fisher Scientific, Illkirch, France) before 100 µL of this final solution was transferred into each Sensititre YeastOne^TM^ YO09 (Thermo Fisher Scientific) plate well. MICs were read after 48 h incubation at 35 °C. 

### 2.6. Physiological Analyses

#### 2.6.1. Energy-Dispersive X-ray Spectroscopy (EDX) Analysis

Fresh cultures of the three strains were fixed in 1 mL of 2.5% Glutaraldehyde in 0.1 M Sodium Cacodylate buffer for at least 1 h. A cytospin was performed with 200 μL of the fixed fungal solution and centrifugation at 800 rpm for eight minutes. EDX was achieved with an INCA X-Stream-2 detector (Oxford Instruments) associated with TM4000 Plus scanning electron microscopy and AztecOne software (Oxford Instruments, Pasadena, CA, USA). The chemical mapping was performed blindly and took into consideration all chemical elements. The results for the weights and atomic percentages of chemical elements for each strain were subjected to principal component analysis with the XLSTAT (Addinsoft, Paris, France) software.

#### 2.6.2. Biolog^TM^ Phenotypic Analysis

Biolog’s advanced phenotypic technology was used for the phenotypic analysis. YT MicroPlates^TM^ (Gen III) (Biolog catalogue no. 1005) were used for oxidation tests and carbon-source assimilation. The selected carbon sources can discriminate between the different profiles of fungal phenotypes [25]. The 96-well YT MicroPlate^TM^ contains a patented Redox tetrazolium dye that changes colour when cellular respiration occurs, conferring a metabolic fingerprint. All strains were cultivated on Biolog Universal Yeast^TM^ (BUY) Agar medium (Biolog catalogue no. 70005). Colonies must be fresh and well-developed. Incubation time depends on the genus. For *Coniochaeta* spp., maximal growth was observed after four to six days of incubation. A fungal suspension was prepared by swabbing some conidia into YT Inoculating Fluid^TM^ (Biolog catalogue no. 72501) adjusted to a 47% transmittance level with the Biolog Turbidimeter^TM^ (Biolog catalogue no. 3587). Then, 100 μL of this suspension was pipetted into each YT MicroPlates^TM^ well. The plates were incubated at 26 °C for one week. The results are shown in the form of a heat map generated by XLSTAT.

### 2.7. DNA Extraction and Sequencing

After five days of incubation on SDA GC at 30 °C, DNA was extracted from the fungal colonies with the Qiagen^TM^ Tissue kit after mechanical lysis using the FastPrep^TM^-24 instrument in bead tubes with G2 lysis buffer (provided with the Qiagen^TM^ Tissue kit). The extraction was performed with the EZ1 Advanced XL^TM^ instrument, following the manufacturer’s instructions. 

Four genomic regions were amplified: the rRNA internal transcribed spacers 1 and 2 (ITS1-2), a fragment of the β-tubulin gene (B-tub2), a fragment of the translation elongation factor 1-alpha gene (TEF-1-α), and the D1/D2 domains of the rRNA large subunit (LSU) (Table 1). PCR mixes included 5 μL of DNA extract to 20 μL of mix (12.5 μL ATG (Ampli Taq Gold^TM^ 360 Master Mix, Applied Biosystems), 6 μL sterile water (DNase/RNase-free), 0.75 μL forward/reverse primer) to a total volume of 25 μL/well. The PCR program for all gene amplifications was as follows: an initial denaturation step at 95 °C for 15 min, followed by 39 cycles of: 1 min denaturation at 95 °C, 30 s annealing at 56 °C, 1 min extension at 72 °C, and a final 5 min extension at 72 °C. The amplicons were visualized on 2% agarose gel with Sybr Safe DNA^TM^ gel stain (Invitrogen, Waltham, MA, USA) using the Safe Imager 2.0 Blue-Light Transilluminator^TM^ (Invitrogen). Sequencing was performed on a 3500 Genetic Analyser^TM^ (Applied Biosystems, Inc.). The sequences were assembled and corrected using Chromas Pro 2.0 and analysed using the BLASTn tool with the reference data available from the GenBank database of the National Center for Biotechnology Information (NCBI).

### 2.8. Phylogenetic Analyses

Five phylogenetic trees were built. The first one was a multilocus tree constructed after concatenating the ITS, B-tub2, and D1/D2 nucleotide sequences of the clinical isolate, two reference strains, and sequences of other *Coniochaeta* species obtained from the GenBank database. The four other trees were each built with only one locus and included several *Coniochaeta* species also collected from GenBank database (accession numbers are detailed in Table 2). All sequences were aligned with Muscle (a tool available in MEGA 11 software). *Phialemonium obovatum* CBS 279.76 was used as the outgroup. The maximum-parsimony phylogenetic trees were constructed with the default settings and branch-robustness estimation was tested using 1000 bootstrap replications with the molecular evolutionary genetics analysis (MEGA) software version 11.

## 3. Results

### 3.1. Macroscopic Characterisation

Macroscopic features confirmed the rapid growth time of *Coniochaeta* species on SDA GC medium at an optimal temperature of 25 °C for all species (Figure 1). However, none of the three yeasts was able to grow at 4 and 45 °C. Colonies of the three isolates were initially white to beige, both on the surface and on the reverse. After four to five days of incubation, *Coniochaeta massiliensis* turned light orange to salmon. All colonies were flat and moist. *Coniochaeta hoffmannii* DSM 2693 and the newly isolated yeast (PMML0158) presented a glabrous aspect, while *Coniochaeta mutabilis* DSM 10716 was typified by an aerial mycelial growth. 

### 3.2. Microscopic Characterisation

The microscopic observation of the three strains revealed wide septate hyphae, numerous cylindrical adelophialides (short phialides without septum), discrete phialides with conical tips exhibiting ellipsoidal to cylindrical and rarely curved conidia, and nonseptate with thin and smooth conidial walls (2 to 3 by 6 to 10 μm). Several conidia were observed aggregating on the hyphae’s sides and most often in clusters. Collarettes were only found in both *Coniochaeta massiliensis* and *C. mutabilis*. No chlamydospore was observed (Figure 2).

### 3.3. Antifungal Susceptibility Testing (AFST)

The minimum inhibitory concentration (MIC) values for the three species for all nine antifungal drugs are shown in (Table 3). The MIC endpoints for all strains were determined as the lowest concentrations inhibiting the growth of 90% of the strains and were determined as described in Perdomo et al. (2011), since there are no validated AFST guidelines for this genus. The MICs of AMB, 5-FC, ITC, POS, and VOR were low for the three isolates. The new species of *Coniochaeta* displayed low echinocandin (AND and CAS) MICs and a high FL MIC, while *C. hoffmannii* and *C. mutabilis* demonstrated the opposite for these four antifungal drugs.

### 3.4. MALDI-TOF MS Identification

The MALDI-TOF MS identification score (log score < 1.90) was below the limit required for obtaining a good identification. The isolate was identified at the genus level as *Coniochaeta* sp. identification score values were generated from MALDI-TOF MS spectra of the new isolate and the two other reference strains from the DSMZ collection, *Coniochaeta mutabilis* DSM 10716 and *Coniochaeta hoffmannii* DSM 2693, which were collected and have been entered in the MALDI-TOF MS database. 

### 3.5. Physiological Analysis

#### 3.5.1. EDX (Energy-Dispersive X-ray Spectroscopy)

The weights and atomic percentages of chemical elements resulting from the chemical mapping performed on the three species of the *Coniochaeta* genus displayed three distinct profiles. In the principal component analysis (Figure 3), each *Coniochaeta* species was very distant from the others, demonstrating highly heterogeneous chemical profiles. 

#### 3.5.2. Biolog^TM^ System

Biolog^TM^ advanced phenotypic technology was very useful for the phenotypic characterisation. The oxidation and assimilation test results were illustrated using heat maps (Figure 4 and Figure 5). The majority of the substrates were not oxidized/assimilated. Each heat map was quite heterogeneous, and both demonstrated similar findings: *Coniochaeta mutabilis* DSM 10716 appeared more divergent from *Coniochaeta massiliensis* (PMML0158) than from *Coniochaeta hoffmannii* DSM 2693. However, *Coniochaeta massiliensis* (PMML0158) appeared to be closely related to *Coniochaeta hoffmannii* DSM 2693.

### 3.6. DNA Sequencing

The sequences were analysed using the BLASTn tool with the reference data available from the GenBank database of NCBI. The BLASTn of the new isolate showed a percent identity ≤97% for the four gene markers. Notably, the best-match pairwise identity was 96.51% for *C. rhopalochaeta* CBS 109872 with ITS (NR172554.1), 96.41% for *C. deborreae* BE19 001008 with TEF-1a (MW890087.1), 97.45% for *C. deborreae* CBS 147215 with D1/D2 (NR076709.1), and 94.44% for *Cosmospora inonoticola* 9361 with B-tub2 (KU563621.1).

### 3.7. Phylogenetic Analyses

We built phylogenetic trees using MEGA 11 software. The multilocus analysis of the first tree (Figure 6) based on the assembled sequences of the ITS, B-tub2, and D1/D2 genetic regions revealed that the new isolate *Coniochaeta massiliensis* (PMML0158) clustered apart and seemed quite distinct from all other *Coniochaeta* species. Then, we constructed four other trees (Figure 7) with only one locus each, including supplementary *Coniochaeta* and *Colletotrichum* species. Within both ITS and TEF-1a trees, *Coniochaeta massiliensis* PMML0158 clustered with other *Coniochaeta* species. However, the D1/D2 and B-tub2 trees could not clearly position the new species PMML0158 in the *Coniochaeta* or *Colletotrichum* genus. However, the TEF-1a tree (Figure 7D) clearly positioned *Coniochaeta massiliensis* (PMML0158) as a distinct species within the *Coniochaeta* genus, which was unambiguously separated from the *Colletotrichum* genus.

### 3.8. Taxonomy


***Coniochaeta massiliensis* Kabtani J. & Ranque S. sp. nov.**


MycoBank: MB843839

(Figure 2A–L). 

Etymology: Named in honour of Marseille (France), the city where it was isolated.

Diagnosis: Closely similar to the other two *Coniochaeta* species examined, displaying the same flat and moist colonies aspect, as well as the absence of the dematiaceous appearance. However, relying on macroscopic features, it differs from *C. mutabilis* in that it lacks aerial growth. On the other hand, *C. massiliensis* presented the same microscopic structures as other species, such as the presence of several adelophialides, discrete phialides, and cylindrical or curved conidia. In this respect, *C. massiliensis* is closer to *C. mutabilis,* due to the decisive presence of the collarette.

Type: France: Marseille. Human body (abscess of the hand), 15 July 2020. (Holotype IHEM 28559/PMML0158, stored in a metabolically inactive state.) GenBank: OM366153 (ITS), ON000097 (B-tub2), OM640093 (TEF-1a), OM366268 (D1/D2).

Description: the macroscopic features were characterised by a rapid growth time on SDA GC medium at an optimal temperature of 25 °C. However, *Coniochaeta massiliensis* was not able to grow at 4 and 45 °C. Colonies were first white to beige, both on the surface and on the reverse after four to five days of incubation, then turned light orange to salmon. Colonies were flat and moist, with a glabrous aspect. There was no aerial mycelial growth. The microscopic features were characterised by the presence of wide septate hyphae, numerous cylindrical adelophialides (short phialides without septum), discrete phialides with conical tips, exhibiting ellipsoidal to cylindrical and rarely curved conidia, nonseptate with thin and smooth conidial walls (2 to 3 by 6 to 10 μm). Several conidia were observed aggregating on the side of the hyphae and most often in clusters. Collarettes were present, but chlamydospores formation was not observed.

The Biolog^TM^ carbon-source assimilation profile showed that *C. massiliensis* PMML0158 can assimilate different carbon substrates, such as 2-keto-D-gluconic acid, D-gluconic acid, D-ribose, D-xylose, D-glucosamine, D-cellobiose, D-melibiose, Palatinose, Turanose, L-sorbose, and β-methyl-D-glucoside. Based on this phenotypic analysis, *C. massiliensis* PMMFL0158 appears similar to *C. hoffmannii* DSM 2693.

#### Host: Human

Additional specimen examined: *C. Hofmann*: Reference: Country of origin unknown. Treated pine stake, before 8 July 1983. (DSM 2693-ATCC 34158–SP33-4.) GenBank: OM366155 (ITS), ON000099 (B-tub2), OM640095 (TEF-1a), OM366270 (D1/D2).

Additional specimen examined: *C. mutabilis*: Reference: Sweden. Origin of isolation unknown, before 14 June 1996. (DSM 10716-EMPA 573, S24 E.) GenBank: OM366154 (ITS), ON000098 (B-tub2), OM640094 (TEF-1a), OM366269 (D1/D2).

## 4. Discussion

*Coniochaeta mutabilis* and *Coniochaeta hoffmannii* are the most familiar human pathogens of the *Coniochaeta* genus and the most widespread and commonly encountered in human samples and severe infections. The MALDI-TOF MS identifications for these two species were relevant, with log scores >2.0. However, this tool was not able to perform a species identification for the new isolate, which led to a molecular analysis targeting four relevant genes: the internal transcribed spacer (ITS1/ITS2) in the RNA ribosomal small subunit (SSU), a fragment of the translation elongation factor 1-alpha gene (TEF-1-α), a fragment of the β-tubulin gene (B-tub2), and the D1/D2 domains of the ribosomal DNA large subunit (LSU). 

The multilocus phylogenetic tree (Figure 6) constructed with the concatenated sequences of ITS, B-tub2, and D1D2 showed that the type strains *C. mutabilis* CBS 157.44 and *C. hoffmannii* CBS 245.38 clustered within distinct clades and that both were distant from *Coniochaeta massiliensis* (PMML0158). Indeed, this newly isolated yeast was divergent from all other *Coniochaeta* species present in the tree. However, in the single-locus trees for ITS and B-tub2, *Coniochaeta massiliensis* (PMML0158) seemed closer to *Coniochaeta* species than in other single-locus trees for D1/D2 and TEF-1a, where it appeared much more distant from *Coniochaeta* species than the genus *Colletotrichum*.

Based on phylogeny, the new isolate seems divergent from the group’s other known taxa of *Coniochaeta*. de Hoog et al. (2000) described *C. lignaria* as being genetically closely related to *C. hoffmannii*. However, in our phylogenetic tree, these yeasts clustered into two different clades, indicating a large distance between the two species. These findings are in line with those of both Perdomo et al. (2011) and Weber and Begerow (2002), who reported significant differences between these species.

Antifungal susceptibility testing showed relatively low MICs for AMB, 5-FC, ITC, POS, and VOR in all strains. These results are in line with those of Perdomo et al. (2011) but contrast with those of other authors [9,13,20] who reported relatively high MICs for AMB and ITC against *C. hoffmannii* and *C. mutabilis.*

We macroscopically observed a fast-growing phenotype in *Coniochaeta* species. This finding contrasts with those of Ahmad et al. (1985), who reported slow colony growth. All species displayed the same flat and moist colonies. Only *C. mutabilis* additionally displayed low aerial growth, as reported by Drees et al. (2007). The primary colony colour remained unchanged (white/beige to orange/salmon). None of the colonies turned brown/dark in culture even after one month of incubation. This aspect has been well described in *C. hoffmanii*, a species that lacks the dematiaceous aspect, whereas *C. mutabilis* is classified within dematiaceous fungi, which are particularly typified by the presence of melanin in hyphae and conidia cell walls, which is responsible for dark pigment emission. Occasionally, certain *C. mutabilis* species lack the melanin property, as mentioned in Khan et al. (2013), Dress et al. (2007), and Perdomo et al. (2011). This was noted for the *C. mutabilis* DSM 10716 strain. Furthermore, in this study, the three species lacked the dematiaceous aspect.

The most remarkable morphological characteristics of the *Coniochaeta* species that have been described were the presence of several adelophialides, discrete phialides, and cylindrical or curved conidia, in addition to the presence or absence of collarettes and chlamydospores, depending on the species [9,14,18,29]. Most of these specific characteristics were observed in the newly described species (PMML0158). Based on this microscopic analysis, we infer that *Coniochaeta massiliensis* is closer to *C. mutabilis,* due to the decisive presence of a collarette.

Morphological characterisation helped for species differentiation, and physiological analysis was more convincing in distinguishing the three species, with the aim of describing the new one as thoroughly as possible. EDX (energy-dispersive X-ray spectroscopy) and Biolog^TM^ phenotypic technology revealed divergent chemical mapping and carbon-source oxidation/assimilation profiles between the new isolate and the two main species of *Coniochaeta* genus. Moreover, the Biolog^TM^ system findings were more relevant, as they revealed that the physiological profile of *Coniochaeta*
*massiliensis* was closer to *C. hoffmannii* DSM 2693 than to *C. mutabilis* DSM 10716.

In conclusion, the clinical yeast isolate PMML0158 is herein described as *Coniochaeta*
*massiliensis*, a new species that can be easily discriminated from the other species in the *Coniochaeta* genus owing to distinct genomic sequences and chemical and physiological profiles. 

## Figures and Tables

**Figure 1 jof-08-00999-f001:**
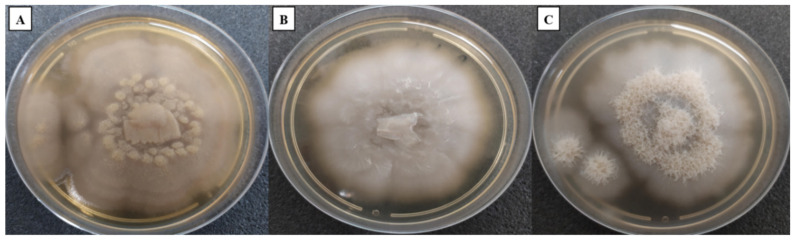
Culture growth on Sabouraud Dextrose Agar + gentamicin and chloramphenicol after five days of incubation at 25 °C. The colour of both the recto and verso of the colonies was white/beige to salmon. (**A**) *Coniochaeta massiliensis* PMML0158. (**B**) *Coniochaeta hoffmannii* DSM 2693. (**C**) *Coniochaeta mutabilis* DSM 10716.

**Figure 2 jof-08-00999-f002:**
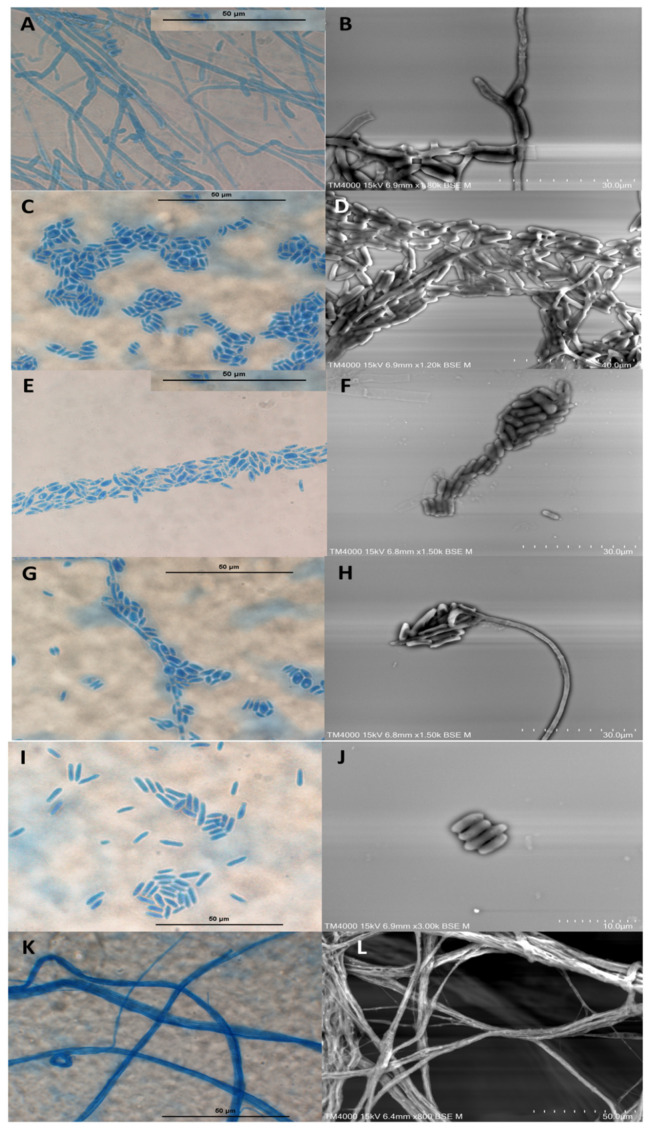
Morphology of *Coniochaeta massiliensis* (PMML0158). (**A**,**B**) Formation of conidiogenous cells (adelophialides) on hyphae and presence of collarette (arrow). (**C**–**F**) Conidia aggregating in clusters. (**G**) Conidia aggregating along the sides of the hyphae. (**H**) Phialoconidia assembled at the phialide tip. (**I**,**J**) Cylindrical conidia with thin and smooth walls. (**K**,**L**) Wide hyphae. Optical microscopy (magnification ×1000). Scale bars: 50 μm. Scanning electron microscopy (15 KeV lens mode 4). Scale bars: **J** = 10 μm; **B**,**F**,**H** = 30 μm; **D** = 40 μm; **L** = 50 μm.

**Figure 3 jof-08-00999-f003:**
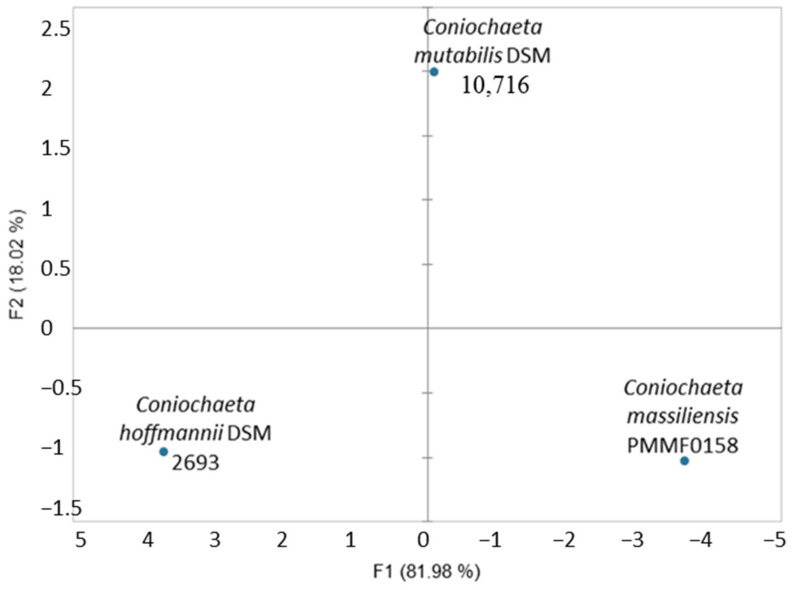
Principal component analysis (PCA) processed with the XLSTAT software of the energy-dispersive X-ray spectroscopy chemical mapping profile, performed for the novel species *Coniochaeta massiliensis* (PMML0158) and two reference strains in the genus. The principal components F1 and F2 explained 100% of the chemical mapping profile variance.

**Figure 4 jof-08-00999-f004:**
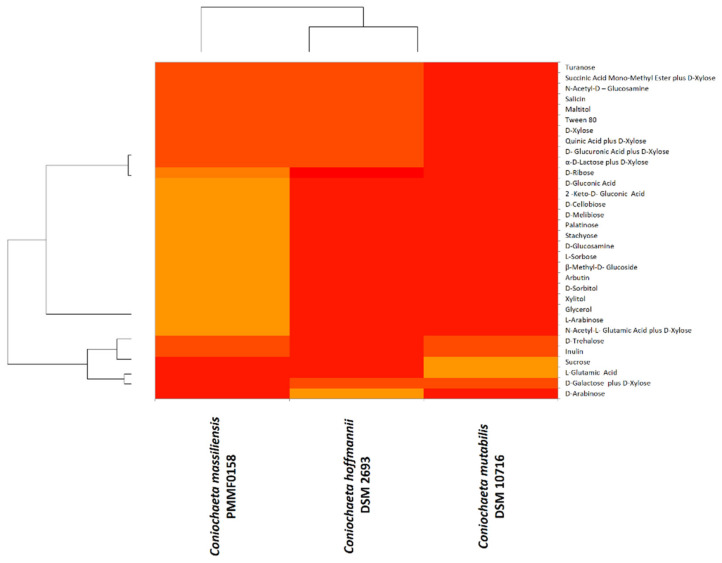
Heat map computed with the XLSTAT software for carbon-source oxidation by the Biolog^TM^ system for the novel species *Coniochaeta massiliensis* (PMML0158) and two reference strains in the genus. Colour-gradient interpretation: the most-oxidized substrates are shown in light orange and the least-oxidized substrates are shown in red.

**Figure 5 jof-08-00999-f005:**
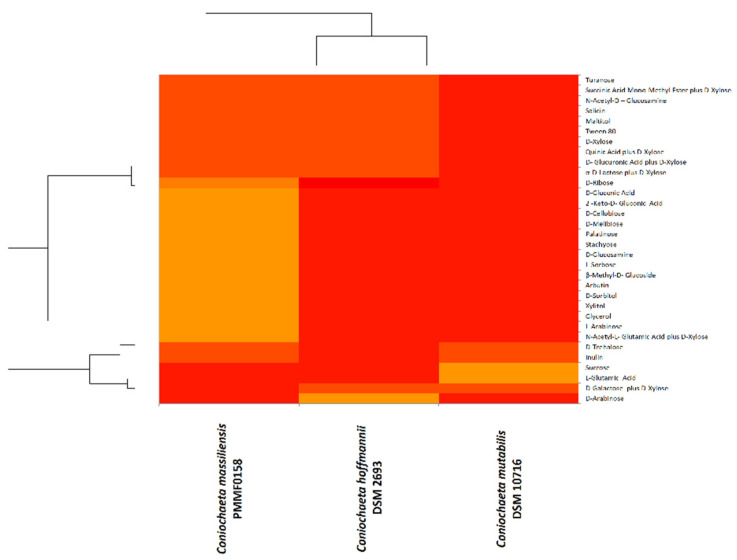
Heat map computed with the XLSTAT software for carbon-source assimilation by the Biolog^TM^ system for the novel species *Coniochaeta massiliensis* (PMML0158) and two reference strains. Colour-gradient interpretation: the most-assimilated substrates are shown in light orange and the least-assimilated substrates are shown in red.

**Figure 6 jof-08-00999-f006:**
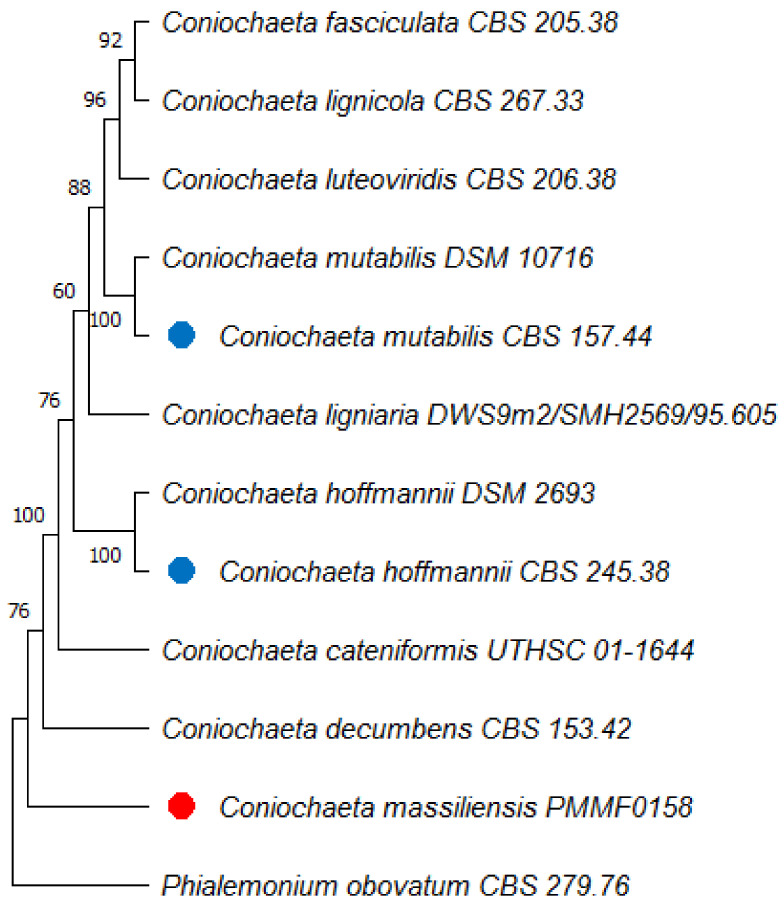
Multilocus phylogenetic tree of the newly isolated species *Coniochaeta massiliensis* PMML0158 (indicated with red dots) and 11 reference strains (type strains are indicated with blue dots), based on the concatenated ITS, B-tub2, and D1/D2 sequences. *Phialemonium obovatum* CBS 279.76 was used as an outgroup. The maximum-parsimony tree was generated using the MEGA 11 software, with 1000-replication bootstrap values.

**Figure 7 jof-08-00999-f007:**
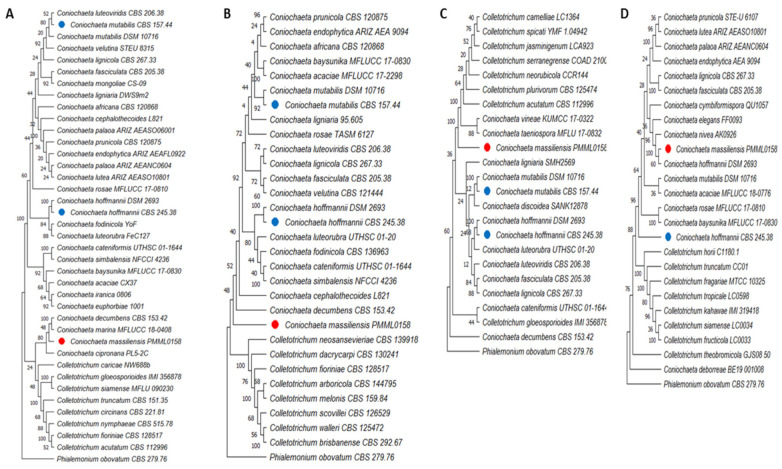
Four single-locus phylogenetic trees using the ITS (**A**), D1/D2 (**B**), B-tub2 (**C**), and TEF-1α (**D**) genomic regions. The species included in each tree differ because the sequences for each locus were not available for all strains. The red dots indicate the new species *Coniochaeta massiliensis* PMML0158 and blue dots indicate type strains. *Phialemonium obovatum* CBS 279.76 was used as the outgroup. The maximum-parsimony tree was generated using the MEGA 11 software, with 1000-replication bootstrap values.

**Table 1 jof-08-00999-t001:** Sets of primers used for amplifying the ITS, B-tub2, TEF1, and D1/D2 genetic regions.

Primers	Sequences	Targeted Regions	References
ITS1	TCCGTAGGTGAACCTGCGG	18S-5.8S	[26]
ITS2	GCTGCGTTCTTCATCGATGC	18S-5.8S	[26]
ITS3	GCATCGATGAAGAACGCAGC	5.8S-28S	[26]
ITS4	TCCTCCGCTTATTGATATGC	5.8S-28S	[26]
ITS1	TCCGTAGGTGAACCTGCGG	18S-5.8S, 5.8S-28S	[26]
ITS4	TCCTCCGCTTATTGATATGC	18S-5.8S, 5.8S-28S	[26]
Bt-2a	GGTAACCAAATCGGTGCTGCTTTC	B-tub2	[27]
Bt-2b	ACCCTCAGTGTAGTGACCCTTGGC	B-tub2	[27]
EF1-728F	CATCGAGAAGTTCGAGAAGG	TEF1	[28]
EF1-986R	TACTTGAAGGAACCCTTACC	TEF1	[28]
D1	AACTTAAGCATATCAATAAGCGGAGGA	28S	[11]
D2	GGT CCG TGT TTC AAG ACG G	28S	[11]

**Table 2 jof-08-00999-t002:** GenBank accession numbers of the reference strains used for the phylogenetic analyses.

Species	Collection ID	GenBank Accession Numbers			
		ITS	B-tub2	D1/D2	TEF1
*Coniochaeta massiliensis*	PMML0158	OM366153	ON000097	OM366268	OM640093
*Coniochaeta mutabilis*	DSM 10716	OM366154	ON000098	OM366269	OM640094
*Coniochaeta hoffmannii*	DSM 2693	OM366155	ON000099	OM366270	OM640095
*Coniochaeta fasciculata*	CBS 205.38	HE610336	HE610350	FR691988	MK693152
*Coniochaeta lignicola*	CBS 267.33	HE610335	HE610353	FR691986	MK693154
*Coniochaeta luteoviridis*	CBS 206.38	HE610333	HE610351	FR691987	NA *
*Coniochaeta hoffmannii*	CBS 245.38	HE610332	HE610352	FR691982	MK693150
*Coniochaeta mutabilis*	CBS 157.44	HE610334	HE610349	AF353604	NA
*Coniochaeta lignaria*	DWS9m2/SMH2569/95.605	KJ188673	AY780113	AF353584	NA
*Coniochaeta cateniformis*	UTHSC 01-1644	HE610331	HE610347	HE610329	NA
*Coniochaeta decumbens*	CBS 153.42	HE610337	HE610348	HE610463	NA
*Coniochaeta velutina*	CBS 121444	NA	NA	GQ154605	NA
*Coniochaeta velutina*	STEU 8315	KY312638	NA	NA	NA
*Coniochaeta simbalensis*	NFCCI 4236	MG825743	NA	NG_068555	NA
*Coniochaeta fodinicola*	YoF/CBS 136963	JQ904607	NA	NG_064287	NA
*Coniochaeta prunicola*	CBS 120875	GQ154540	NA	NG_066151	NA
*Coniochaeta prunicola*	STE-U 6107	NA *	NA	NA	MK693162
*Coniochaeta africana*	CBS 120868	GQ154539	NA	NG_066150	NA
*Coniochaeta palaoa*	ARIZ AEASO06001	MZ241149	NA	NA	NA
*Coniochaeta palaoa*	ARIZ AEANC0604	MK458764	NA	NA	MZ241188
*Coniochaeta marina*	MFLUCC 18-0408	MF422164	NA	NA	NA
*Coniochaeta cipronana*	PL5-2C	MG828883	NA	NA	NA
*Coniochaeta rosae*	MFLUCC/17-0810	MG828880	NA	NA	MG829197
*Coniochaeta rosae*	TASM 6127	NA	NA	NG_066204	NA
*Coniochaeta baysunika*	MFLUCC/17-0830	MW750761	NA	MG828996	MG829196
*Coniochaeta acaciae*	CX37	MW750756	NA	NA	NA
*Coniochaeta acaciae*	MFLUCC 18-0776	NA	NA	NA	MT503199
*Coniochaeta acaciae*	MFLUCC 17-2298	NA	NA	MG062737	NA
*Coniochaeta fibrosae*	CX04D1	MW077645	NA	NA	NA
*Coniochaeta mongoliae*	CS-09	KP941078	NA	NA	NA
*Coniochaeta iranica*	806	KP941076	NA	NA	NA
*Coniochaeta euphorbiae*	1001	KY064029	NA	NA	NA
*Coniochaeta cephalothecoides*	L821	MW447035	NA	KY064030	NA
*Coniochaeta luteorubra*	FeC127	MZ241160	NA	NA	NA
*Coniochaeta luteorubra*	UTHSC 01-20	NA	HE610346	HE610328	NA
*Coniochaeta lutea*	AEASO10801	MZ241150	NA	NA	MZ241193
*Coniochaeta endophytica*	ARIZ AEAFL0922	MZ241147	NA	NA	NA
*Coniochaeta endophytica*	ARIZ AEA 9094	NA	NA	NG_075158	NA
*Coniochaeta vineae*	KUMCC 17-0322	NA	MN485898	NA	NA
*Coniochaeta taeniospora*	MFLU 17-0832	NA	MN509784	NA	NA
*Coniochaeta discoidea*	SANK12878	NA	AY780134	NA	NA
*Coniochaeta elegans*	FF0093	NA	NA	NA	MZ267815
*Coniochaeta nivea*	AK0926	NA	NA	NA	MZ267793
*Coniochaeta deborreae*	BE19_001008	NA	NA	NA	MW890087
*Coniochaeta cymbiformispora*	QU1057	NA	NA	NA	MZ267839
*Coniochaeta endophytica*	AEA 9094	NA	NA	NG_075158	MK693159
*Colletotrichum nymphaeae*	CBS 515.78	NR_111736	NA	NA	NA
*Colletotrichum caricae*	NW688b	NR_111736	NA	NA	NA
*Colletotrichum fioriniae*	CBS 151.35	NR_111458	NA	NG_069002	NA
*Colletotrichum circinans*	CBS 128517	NR_111747	NA	NA	NA
*Colletotrichum gloeosporioides*	CBS 221.81	NR_111457	NA	NA	NA
*Colletotrichum gloeosporioides*	IMI 356878	NA	AJ409291	NA	NA
*Colletotrichum acutatum*	IMI 356878	NR_160754	NA	NA	NA
*Colletotrichum acutatum*	CBS 112996	NA	AJ409296	NA	NA
*Colletotrichum siamense*	CBS 112996	NR_144794	NA	NA	NA
*Colletotrichum siamense*	LC0034	NA	NA	NA	JQ071904
*Colletotrichum truncatum*	MFLU 090230	NR_144784	NA	NA	NA
*Colletotrichum truncatum*	CC01	NA	NA	NA	MW030430
*Colletotrichum plurivorum*	CBS 125474	NA	MG600985	NA	NA
*Colletotrichum jasminigenum*	LCA923	NA	HM153770	NA	NA
*Colletotrichum serranegrense*	COAD 2100	NA	KY407896	NA	NA
*Colletotrichum neorubicola*	CCR144	NA	MN186400	NA	NA
*Colletotrichum spicati*	YMF 1.04942	NA	OL981226	NA	NA
*Colletotrichum fructicola*	LC0033	NA	NA	NA	JQ071903
*Colletotrichum theobromicola*	GJS08_50	NA	NA	NA	GU994506
*Colletotrichum horii*	C1180.1	NA	NA	NA	GQ329693
*Colletotrichum tropicale*	LC0598	NA	NA	NA	JQ071909
*Colletotrichum kahawae*	IMI 319418	NA	NA	NA	JQ071908
*Colletotrichum fragariae*	MTCC 10325	NA	NA	NA	JQ071906
*Colletotrichum camelliae*	LC1364	NA	JN936976	NA	NA
*Colletotrichum dacrycarpi*	CBS 130241	NA	NA	NG_073638	NA
*Colletotrichum neosansevieriae*	CBS 139918	NA	NA	NG_070628	NA
*Colletotrichum arboricola*	CBS 144795	NA	NA	NG_070064	NA
*Colletotrichum scovillei*	CBS 126529	NA	NA	NG_070041	NA
*Colletotrichum walleri*	CBS 125472	NA	NA	NG_070040	NA
*Colletotrichum melonis*	CBS 159.84	NA	NA	NG_070037	NA
*Colletotrichum brisbanense*	CBS 292.67	NA	NA	NG_070034	NA
*Phialemonium obovatum*	CBS 279.76	HE610365	HE599334	FR691997	LT634003

* NA, not available.

**Table 3 jof-08-00999-t003:** Results of in vitro antifungal susceptibility testing.

MIC * (mg/L) Read at 48 h									
	AMB	AND	CAS	5-FC	FL	ITC	MIF	POS	VOR **
*Coniochaeta massiliensis* PMML0158	0.25	0.5	1	2	8	0.25	1	0.25	0.12
*Coniochaeta hoffmannii* DSM 2693	0.12	4	4	1	4	0.06	>32	0.12	0.12
*Coniochaeta mutabilis* DSM 10716	0.12	4	8	1	2	0.015	>32	0.03	0.03

* MIC, minimum inhibitory concentration; ** AMB, amphotericin B; AND, anidulafungin; CAS, caspofungin; 5-FC, 5-fluorocytosine; FL, fluconazole; ITC, itraconazole; MIF, micafungin; POS, posaconazole; VOR, voriconazole.

## Data Availability

The *Coniochaeta massiliensis* holotype is available in the IHU MI (no. PMML0158) and IHEM (no. 28559) strain collections. The nucleotide sequences are available in GenBank (accession nos. OM366153, ON000097, OM640093, and OM366268). The datasets analysed in the current study are available from the corresponding author upon reasonable request.

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
