# Peer review of "Coniochaeta massiliensis sp. nov. Isolated from a Clinical Sampl28"

_jof, 2022, doi:10.3390/jof8100999_

Round 1
Reviewer 1 Report
The article “Description of Coniochaeta massiliensis a novel Coniochaeta species isolated from a clinical sample” describes a novel fungus best summed up by the authors by “Species identification and thorough description were based on apposite and reliable phylogenetic and phenotypic approaches. The phylogenetic methods included a multilocus nucleotide analysis of four genomic regions: ITS (rRNA internal transcribed spacers 1 and 2), TEF-1α (translation elongation factor-1alpha), TUB2 (β-tubulin2), and D1/D2 domains (28S LSU rRNA).”. While not my field, the phenotypic description and experimental approaches for deriving the phenotype appear to be reasonably extensive. My major problem with the paper is that with the advent of inexpensive sequencing phenotypic descriptions are much less important than genotypic information for identification and clinical diagnosis of a species. The authors use partial sequences from a set of marker genes to determine phylogenetic placement and the assertion of a novel species. I’m not sure if these marker gene sequences are appropriate for the given genus but they seem reasonable. There should be more of an explanation for why these particular marker gene sequences were chosen. The paper should also be explicit about the pairwise percent identity of the marker gene sequences of the novel species to other species and why this indicates a novel species. The authors should justify why they did not attempt to generate a high quality draft genome sequence. Lastly, the authors need to further justify their choice of species to include in the phylogenetic analysis – presumably the primary reason was the availability of the marker genes sequences for those species. What references were used to determine the set of species in the Coniochaeta genus? From the paper, “Therefore, the six species of the Lecythophora genus (L. lignicola, L decumbens, L. fasciculata, L. luteoviridis, L. mutabilis and L. hoffmannii) were transferred to the Coniochaeta genus [9, 10].”, and “. The genotypic characteristics of these three strains were also compared with those of seven other type strains in the genus Coniochaeta, including: C. fasciculate CBS 205.38, C. lignicola CBS 267.33, C. luteoviridis CBS 206.38, C. hoffmannii CBS 245.38, C. mutabilis CBS 157.44, C. lignaria DWS9m2/SMH2569/95.605, and C. cateniformis UTHSC 01-1644.”. Are these the only eight species in the genus? No – reference 9 above mentions Coniochaeta polymorpha for example. Since C. decumbens and other Coniochaeta species were not included in the phylogenetic analysis how do we know they are not the same species as Coniochaeta PMML0158. The paper states and figure 6 supports, “The multilocus analysis of the tree strains (Figure 6) revealed the presence of 2 main clades; the first one is divided into five subclades. The second one includes the newly isolated strain of Coniochaeta PMML0158, which appears distinct from all other Coniochaeta species.”. The fact that the novel species is in a clade by itself supports it being a novel species but raises the question whether it has been assigned to the correct genus. What are the other closest genera to Coniochaeta and why weren’t species from those genera included in the phylogenetic analysis? When I looked in Genbank for Coniochaeta genome sequences I found C. prunicola, C. pulveracea, and nine unidentified Coniochaeta species. Why weren’t the marker gene sequences extracted from these genomes and included in the analysis?
A small issue: “Colonies of the three isolates were initially white to beige, both on the surface and reverse. After four to five days of incubation, Coniochaeta massiliensis turned light orange to salmon. All colonies were flat and moist. Coniochaeta hoffmannii DSM 2693 and the newly isolated yeast (PMML0158) presented a glabrous aspect, while Coniochaeta mutabilis DSM 10716 was typified by an aerial mycelium growth.” Are Coniochaeta massiliensis and the newly isolated yeast (PMML0158) the same? Yes – confusing to refer to this specie/sample in multiple ways.
Author Response
Question 1:
My major problem with the paper is that with the advent of inexpensive sequencing phenotypic descriptions are much less important than genotypic information for identification and clinical diagnosis of a species. The authors use partial sequences from a set of marker genes to determine phylogenetic placement and the assertion of a novel species. I’m not sure if these marker gene sequences are appropriate for the given genus but they seem reasonable. There should be more of an explanation for why these particular marker gene sequences were chosen.
Answer 1:
We choose these genes because they are the most relevant for fungi’s identification (Irinyi et al. 2015. Medical mycology 53 (4), 313-337). The ITS and D1-D2 regions of the rRNA gene are the gold standard Fungal genomic Barcodes. Analyzing the BTUB and TEF1 alpha regions add even more precision in particular fungal clades
Question 2:
The paper should also be explicit about the pairwise percent identity of the marker gene sequences of the novel species to other species and why this indicates a novel species.
Answer 2:
The NCBI Blast of the new isolate showed a percent identity ≤97 % for the four selected loci, while the threshold for discriminating species is < 99%.
Below is a table with the highest obtained pairwise percent identity of each of the four loci.
|
ITS |
Identity |
TEF-1a |
Identity |
B-tub2 |
identity |
D1D2 |
identity |
Coniochaeta massiliensis |
Coniochaeta rhopalochaeta |
96.51% |
Coniochaeta deborreae |
96.41% |
Cosmospora inonoticola |
94.44% |
Coniochaeta deborreae |
97.45% |
Question 3:
The authors should justify why they did not attempt to generate a high quality draft genome sequence.
Answer 3:
We did not sequence the whole genome of our isolate because this approach has not been fully validated for fungal species identification to date. Indeed, the number of fungal (especially Coniochaeta spp.) genomes is rather low, and the quality of the contigs remains questionable. We preferred to use a polyphasic concept combining loci that are validated by the international mycologists’ community. These accurate genotypic and phenotypic approaches have been used and published in 2021 by Arnold et al. (doi: 10.1099/ijsem.0.005003. PMID: 34731078) for the description of three new Coniochaeta species based on phylogenetic analyses of only two loci (ITS rDNA and translation elongation factor 1-alpha).
Question 4:
Lastly, the authors need to further justify their choice of species to include in the phylogenetic analysis – presumably the primary reason was the availability of the marker genes sequences for those species. What references were used to determine the set of species in the Coniochaeta genus? From the paper, “Therefore, the six species of the Lecythophora genus (L. lignicola, L decumbens, L. fasciculata, L. luteoviridis, L. mutabilis and L. hoffmannii) were transferred to the Coniochaeta genus [9, 10].”, and “. The genotypic characteristics of these three strains were also compared with those of seven other type strains in the genus Coniochaeta, including: C. fasciculate CBS 205.38, C. lignicola CBS 267.33, C. luteoviridis CBS 206.38, C. hoffmannii CBS 245.38, C. mutabilis CBS 157.44, C. lignaria DWS9m2/SMH2569/95.605, and C. cateniformis UTHSC 01-1644.”. Are these the only eight species in the genus? No – reference 9 above mentions Coniochaeta polymorpha for example. Since C. decumbens and other Coniochaeta species were not included in the phylogenetic analysis how do we know they are not the same species as Coniochaeta PMML0158.
Answer 4:
As hypothesized by the reviewer, we included the species for which complete multilocus data were available for the phylogenetic analyses. We are confident that Coniochaeta PMML0158 is a new specie among Coniochaetaceae since none of the available sequences in GenBank achieves >98% identity. As suggested by the reviewer, we compared the sequences of Coniochaeta PMML0158 with all Coniochaetaceae whole-genome shotgun contigs in GenBank. The identity scores were < 96% for each locus (<94% for ITS) indicating that Coniochaeta PMML0158 sequences are significantly different from all other deposited Coniochaetaceae genomic sequences. To answer the reviewer’s query more precisely: the ITS identity scores were 93% for C. decumbens, and 91% for C. polymorpha.
Question 5:
The paper states and figure 6 supports, “The multilocus analysis of the tree strains (Figure 6) revealed the presence of 2 main clades; the first one is divided into five subclades. The second one includes the newly isolated strain of Coniochaeta PMML0158, which appears distinct from all other Coniochaeta species.” The fact that the novel species is in a clade by itself supports it being a novel species but raises the question whether it has been assigned to the correct genus. What are the other closest genera to Coniochaeta and why weren’t species from those genera included in the phylogenetic analysis? When I looked in Genbank for Coniochaeta genome sequences I found C. prunicola, C. pulveracea, and nine unidentified Coniochaeta species. Why weren’t the marker gene sequences extracted from these genomes and included in the analysis?
Answer 5:
Please refer to the data in our response to Question 2. The >95% identity scores obtained of each loci support doubtlessly the fact that our isolate belongs to the Coniochaeta genus. The species in the genus that were not-included in the phylogenetic analyses lacked sequences for the loci we used. But single locus BLAST analysis yielded <98% identity in each case.
Question 6:
A small issue: “Colonies of the three isolates were initially white to beige, both on the surface and reverse. After four to five days of incubation, Coniochaeta massiliensis turned light orange to salmon. All colonies were flat and moist. Coniochaeta hoffmannii DSM 2693 and the newly isolated yeast (PMML0158) presented a glabrous aspect, while Coniochaeta mutabilis DSM 10716 was typified by an aerial mycelium growth.” Are Coniochaeta massiliensis and the newly isolated yeast (PMML0158) the same? Yes – confusing to refer to this specie/sample in multiple ways.
Answer 6:
We agree with the reviewer’s remark; indeed, Coniochaeta massiliensis is the newly isolated yeast (PMML0158). We homogenized the text for clarification.
Reviewer 2 Report
This manuscript "Description of Coniochaeta massiliensis a novel Coniochaeta species isolated from a clinical sample" is an interesting piece of work worth to be considered in JOF but it needs major improvements. Authors are advised to do the phylogenetic analyses properly as the present phylogenetic analyses are simple. Please see other annotated comments in the PDF.

Author Response
We enhanced the phylogenetic analyses as required by the reviewer.
We took into account all other comments that were annotated in the PDF.
Round 2
Reviewer 1 Report
The authors' responses to my questions were fine. I did not see that they incorporated any of those responses into the article however based on the markup. I could not find the phylogenetic results section - it appeared that it may somehow have been hidden behind figure 4?
Author Response
In agree with the reviewer’s comments, we modified the format of figure 4 for clarification. The phylogenetic analysis is detailed in section 3.6 and Figure 6. For whom is interested, any further analyses is possible because all our nucleotide sequences have been deposited and ca be accessed in GenBank.
Reviewer 2 Report
This manuscript "Description of Coniochaeta massiliensis a novel Coniochaeta species isolated from a clinical sample" is revised well but it needs more revisions before it is accepted. Please see the annotated comments.
Authors are advised to italicize the scientific names.

Author Response
In agree with the reviewer’s comments, we took into account all annotated comment, and we italicized all scientific names.